# Functional Therapeutic Strategies Used in Different Stages of Alzheimer’s Disease—A Systematic Review

**DOI:** 10.3390/ijerph191811769

**Published:** 2022-09-18

**Authors:** Anna Olczak, Aleksandra Truszczyńska-Baszak, Adam Stępień, Krzysztof Górecki

**Affiliations:** 1Military Institute of Medicine, Rehabilitation Clinic, 128 Szaserów Street, 04-141 Warsaw, Poland; 2Faculty of Rehabilitation, Józef Piłsudski University of Physical Education in Warsaw, 00-968 Warsaw, Poland; 3Military Institute of Medicine, Neurological Clinic, 128 Szaserów Street, 04-141 Warsaw, Poland; 4Military Institute of Medicine, Cardiac Surgery Clinic, 128 Szaserów Street, 04-141 Warsaw, Poland

**Keywords:** Alzheimer’s disease, functional therapy, movement exercises, neurorehabilitation

## Abstract

As Alzheimer’s disease develops, the central nervous system is gradually damaged. It is manifested by progressive dementia and the appearance of neurological and extrapyramidal symptoms that impair everyday functioning. The aim of the study was to evaluate the influence of physical exercise on cognitive and motor functions in various stages of Alzheimer’s disease. Methods: Four databases (PubMed, Scopus, Ovid, and Cochrane Library) were searched for relevant papers published between 2012 and May 2022. The works were assessed in terms of the adopted inclusion criteria. The measures of the results were changed in the parameters assessing motor and cognitive functions. Methodological quality was assessed using the Cochrane Collaboration. This review was recorded with the Cochrane Library: CRD42022340496. The results of the database search showed 302 articles, 12 of which were included in the review. All studies have shown a significant positive effect on improving cognitive and motor functions. This systematic review revealed a beneficial effect in improving cognitive and motor functions after the application of various kinds of activities, especially in the early and mild stages of Alzheimer’s disease.

## 1. Introduction

Alzheimer’s disease (AD) is a progressive degenerative disease of the nervous system. It is the most common (50–60%) cause of dementia in people over 65 years old [1,2].

It causes the accumulation of proteins with a pathological structure in the brain, which leads to the death of nerve cells. As a consequence, the amount of neurotransmitters necessary for the proper functioning of the brain is reduced [3,4].

In the world, 15–21 million people suffer from Alzheimer’s disease [5].

There are no specific causes for Alzheimer’s disease. In about 5% of cases, there is a genetic condition, while in about 95% of cases, Alzheimer’s disease is not related to heredity. The risk factors include old age, female gender, arterial hypertension, type 2 diabetes, increased concentration of the amino acid-homocysteine in the blood, previous head injuries, heart failure, and lack of social contact [6].

As Alzheimer’s disease progresses, the central nervous system is gradually damaged, resulting in various symptoms. However, the main symptom is progressive dementia, which may be accompanied by neurological symptoms such as extrapyramidal symptoms, manifested by the so-called “Parkinsonian syndrome”. Moreover, this syndrome may appear as a result of side effects of certain drugs (neuroleptics), which are often necessary to administer to patients with Alzheimer’s disease [3,6].

Depending on the severity of symptoms, there are three stages of Alzheimer’s disease.

The initial stage—in which there is a slight change in behavior, a reduction in the quality of mental processes, and mild cognitive impairment [7].

Moderate stage in which behavioral disturbances increase including inter alia, wandering, aimlessly walking, restlessness, outbursts of anger, making the same movements, and aggression towards the caregiver. Delusions may arise, such as robbing, cheating, stalking, and hallucinating. It may be difficult for the patient to recognize family members. Symptoms of Alzheimer’s disease at this stage may also include excessive appetite, chewing various objects, disturbances in the rhythm of sleep and wakefulness (increased nighttime activity, naps during the day), and problems with independent eating, dressing, and bathing [8,9].

An advanced stage in which, in addition to the symptoms mentioned above, neuropsychiatric symptoms such as depression, anxiety or apathy may appear. In addition, during the period in question, the symptoms of Alzheimer’s disease are accompanied by movement disorders (e.g., motor slowdown—bradykinesia, increased muscle tone), and urinary and/or stool incontinence. Often the patient spends more and more time in bed until he is completely immobilized in the final stages of the disease. There is also a loss of the ability to communicate and recognize people and even oneself. Help in all activities becomes necessary [7].

In order to improve the cognitive state and slow down the clinical symptoms of Alzheimer’s disease, it is recommended—in addition to pharmacological treatment—psychological treatment, especially cognitive therapy (training memory, orientation in reality) and reminiscence therapy (evoking memories with the use of appropriate stimulating stimuli, e.g., photography, letters, souvenirs). In addition, patients often participate in occupational therapy and community therapy in order to create a safe and friendly environment for the patient. More and more space is devoted to other forms of therapy such as music therapy, physical therapy, or exercise therapy [10,11,12,13]. Researchers have recognized the importance of these therapies in improving the functioning of the daily life of Alzheimer’s patients.

However, research works concern patients with a certain level of cognitive and/or motor impairment.

The aim of our study was to evaluate the influence of functional exercise on cognitive and motor functions in various stages of Alzheimer’s disease.

## 2. Materials and Methods

Standard databases were searched to gather material for analysis. Subsequently, the selected screenings were verified according to the search strategy as well as the inclusion and exclusion criteria. Works meeting the assumed criteria were assessed using Cochrane [14].

The present study was registered under CRD42022340496 in the International Prospective Register of Systematic Reviews (PROSPERO).

### 2.1. Search Strategy

A comprehensive, systematic search was performed using the four databases of PubMed, Scopus, Ovid, and the Cochrane Library. The search was limited to full-text works published in the last 10 years, from 2012 to May 2022. In order to find the relevant research, the following keywords were used: Alzheimer’s disease, functional therapy, and movement exercises. Inclusion and exclusion criteria have been established.

Inclusion criteria: (1) patients diagnosed with Alzheimer’s disease aged 60 and above; (2) different stages of the disease; (3) therapeutic intervention in the form of physical exercises and various forms of music therapy; (4) randomized controlled trials, quasi-experimental trials, cross-controlled trials, or controlled trials without randomization; (5) measurement of at least one parameter related to motor and cognitive functions before and after the intervention; (6) full-text articles in English; (7) human research; (8) test results independent of gender.

Exclusion criteria: (1) other disease entity; (2) age under 60; (3) individual case studies, review articles, short messages, letters with insufficient information for the analysis of results, guidelines, theses, dissertations, qualitative research, abstracts from scientific conferences or animal research; (4) therapy other than function exercise therapy.

### 2.2. Study Selection and Data Extraction

The search was applied to four electronic databases. Then all found articles were checked in terms of titles and in terms of repetition. After the duplicates were removed, all articles were checked against inclusion and exclusion criteria to find suitable articles for analysis. In the third stage, the articles were read in full and a final selection was made. The next step was to conduct a detailed analysis of each study. The assessment of the methodological quality of the study and the risk of bias was based on Cochrane and the selection of important points responsible for good methodological quality. Therefore, the following information was analyzed: randomization of the sample (or lack thereof); generating a random string; blindness of evaluators; similar group in the initial assessment; criteria for the inclusion of participants; description of the experimental protocol; statistical comparison between groups; description of the sample at startup and description of the results.

## 3. Results

A total of 302 full-text articles were found in the searched databases. As a result of adjusting the settings in the databases, the systems eliminated 92 works. On the other hand, the analysis of the titles and authors of the works showed duplicates in the amount of 79 items, which were also rejected. In addition, non-compliance with the inclusion criteria was noted, which led to the exclusion of another 80 entries. 51 papers were identified, from which, after a preliminary analysis of abstracts or the content of the paper, another 25 papers were excluded.

For further analysis, 26 papers were initially qualified, potentially meeting the criteria for eligibility for analysis under the systematic review. Reading the articles revealed an additional 14 articles in which the procedure did not meet the inclusion criteria. Finally, 12 articles were qualified for the evaluation.

The search strategy is presented in Figure 1.

### 3.1. Characteristics of the Studies

The methodological quality of the studies included in the analysis (Table 3) is high in most of the studied cases.

The characteristics of the articles included are presented in Table 1 and Table 2.

### 3.2. Evaluation of the Methodological Quality of the Included Research

The Cochrane Collaboration tool was used for the methodological evaluation of the analyzed works [27,28].

All of the works assessed had predetermined assumptions and were characterized by clear inclusion and exclusion criteria, which were defined at the beginning. The methodological quality assessment is presented in Table 3.

Percentage analysis was performed to assess the risk of bias in articles according to the criteria mentioned above. The presence of these criteria is important as they contribute to making studies less prone to bias or systematic errors that may compromise the accuracy of the scientific evidence. Regarding sample randomization, an important methodological procedure was performed in all 100% of the articles. Data on random sequence generation was presented in only three articles of 25% of the studies. The blindness of the evaluators was described in 7 of the discussed articles (58%). In each of the 12 articles, the groups were shown to be similar at the starting point of the intervention (100%), confirmed by statistical comparison between the groups. All articles presented and described the inclusion criteria for participants and a description of the results. However, the description of the experimental protocol did not appear in one of the analyzed articles (92%). Similarly, the description of sample losses was presented in 6 out of 12 articles (50%). It was therefore observed that out of the nine criteria assessed in the study, compliance was low in one (random sequence generation), mean in two (blinding of evaluators and description of sample losses), and near the maximum score in two studies (description of the experimental protocol and statistical comparison among groups). The other four met the requirements 100%. Thus, the methodological quality of the analyzed articles was very good and good. The analysis of the articles selected for this review, included in the percentage of points assessed, reduced the risk of bias.

## 4. Discussion

The purpose of this systematic review was to evaluate the effects of functional exercises on cognitive and motor functions at different stages of Alzheimer’s disease.

For the methodological evaluation of the analyzed works, the Cochrane Collaboration tool, also used by other researchers, was used [27,28].

In the analyzed literature from the ten-year period, twelve articles met the criteria. They examined the effect of functional exercise on cognitive and motor functions, and all of them demonstrated the positive impact of various forms of activities on the functions studied. The aim of this work was to test the influence of various forms of activity on cognitive [24,26,29] and motor functions [15,16,17,18,20,21,22,23] or to compare the effects of a functional exercise program with cognitive training [19,25]. Only two of the studies reported the results, mainly with regard to cognitive functions [24,26]. All studies were carried out on a group of people over 60 years of age.

In addition, all studies were conducted on a group of patients in the mild to moderate and severe stages of the disease. Most studies concerned the early stage of the disease [15,17] and the moderate stage of the disease [16,18,19,20,22,25,26]. Advanced stage research is extremely rare. there is only one such study in our review [25].

In the analysis of the above-selected studies, the positive effect of various forms of physical activity in patients with Alzheimer’s disease was demonstrated [15,16,17,18,19,20,21,22,23,24,25]. Other forms of activating people with cognitive disorders may also bring positive effects in inhibiting their progression [18,21,23,26]. It should be emphasized that this relationship is noticeable and repeated in various studies. Improving the functioning of the respiratory and circulatory system directly improves functional abilities and indirectly improves cognitive abilities [15,16,17,24,25,26].

Our survey shows that most studies that the observed improvements in cognitive and motor functions in patients with Alzheimer’s disease were recorded in the early and moderate stages.

It should be noted that Alzheimer’s disease in its early stages requires observation and appropriate diagnosis to distinguish it from dementia. For this reason, studies on dementia where Alzheimer’s disease was a component were included in the analyzed studies [20,21].

Distinctive results are studies on aerobic exercise, functional effects, and cognitive abilities in people in the early stages of Alzheimer’s disease. In addition, the impact of exercises with music, exercises using dogs for therapy, and exercises originating in Asian countries, i.e., Tai Chi or the game of Go.

The 26-week aerobic exercise program, with increasing duration from 60 to 150 min per week, resulted in a 1% increase in performance in the area of complex activities of daily living. On the other hand, in the group administered with other forms of physical activity, there was an 8% decrease in the same area after 26 weeks [17].

Similar results were reported from another improvement program, which was a bit shorter, at16 weeks. Participants performed the exercises for 60 min three times a week. It should be noted that in this rehabilitation program, from the first to the fourth week, participants performed strength exercises to strengthen their lower limbs. For the next 12 weeks, they performed aerobic exercises on a treadmill, a cyclo-ergometer, or conducted cross-training classes. In this study, highly specialized methods in the form of VO2 peak measurement were used to assess fitness, while the Symbol Digit Modalities Test (SDMT) was used for neuropsychiatric assessment [16].

Also, the American research conducted by Fang Yu and Ruth M. Swartwood showed a positive effect of aerobic exercise in patients with Alzheimer’s disease. During the 6-month program, participants with mild to moderate AD performed a cycling training program. The cycling workouts had a progressive load over the duration of the study. The target load was 45 min of cycling three times a week. The moderate training load was individually selected for the participant using the heart rate reserve at 65–75% HR and the Borg rating of perceived exertion scale, at the level of 12–14. The initial load was selected individually in such a way that the participant would perform a 10-min ride without interruption. When the subject was able to perform 3, 10-min sessions in a row, the load was increased by 5% in the heart rate reserve and by 1 in the Borg rating of perceived exertion scale. The process of increasing the load continued until the intended moderate-load parameters were achieved. Classes were held at the YMCA gym, or on senior campuses. Measurements of cognitive and physical function, and the Alzheimer’s disease triad, i.e., cognitive impairment, functional decline, and behavioral and psychological symptoms of dementia (BPSD) were taken at the start of the program and at 3 and 6 months. As part of the study, there were four group meetings where the participants were interviewed about feelings, memory changes, and changes in their daily life. These meetings were also attended by caregivers of AD patients, who also answered questions. Observations from the study by Fang Yu and Ruth M. Swartwood proved that adapted and pleasant aerobic exercise in AD patients had a positive effect on the level of the general functioning of these patients, although it did not improve their cognitive abilities [22].

Dog therapy, included in one of the above-selected studies, differs from the standard method of activating patients with Alzheimer’s disease. The therapeutic program in this study was 8 months, during which dog therapy sessions were administered once a week for 45 min. Patients were assessed using the Mini Examination Cognitive, a modified Bartell Index, the Cornell Scale for Depression in Dementia, and the Neuropsychiatric Inventory. Patients were assessed at the beginning, during, and at the end of the dog-therapy cycle. The basic element in the first therapeutic sessions was to build a bond with the dog based on positive associations. Then, along with subsequent therapeutic units, the motor tasks were carried out and they gradually became more and more demanding, increasing the involvement of motor coordination and the patient’s attention. The results of this study showed a positive effect of this type of interaction between the patient and the dog [21].

Other forms of activating the Alzheimer’s patient, such as Tai Chi, should also be appreciated. Among the selected works, one study conducted on volunteers in a country where Tai Chi is a popular form of activity. The time limit for this study was 12 months during which participants were screened before the start of the program at 2, 6, and 12 months. As part of the intervention, an introductory phase was distinguished, which lasted from 4 to 6 weeks, during which participants learned about the correct forms of movement in Tai Chi. The next step was the maintenance phase during which participants exercised at home or at the center. To ensure the uniformity of the exercises, the participants received a video CD. The Tai Chi program assumed training three times a week for 30 min. As part of the overall participant assessment, this study used numerous research tools such as the Clinical Dementia Rating Scale (CDR), Disability Assessment for Dementia (DAD), Cornell Scale for Depression in Dementia (CSDD), Chinese Neuropsychiatric Inventory (NPI), and Berg Balance Scale (BBS). These multicentre studies showed a significant positive effect of Tai Chi on the protection of cognitive abilities and general functioning in the elderly [18].

Interesting results were obtained by Chinese scientists Qiao Lin, Yungpeng Cao, and Jie Gao in their research on the effects of playing Go on patients with AD. In his experiment, the respondents were divided into three groups: the control group of those who did not play Go, those who played Go briefly (1 h of games a day), and those who played Go for long periods (2 h of games a day). All three groups were proportionally similar in terms of age, sex, education time, BMI, alcohol consumption, smoking, and diabetes. None of the participants played Go prior to participating in the study. During the project, participants belonging to research groups had to learn the rules of the game at the beginning and gradually learned to play it under the guidance of a Go player. Numerous tools were used for the research evaluation: Montgomery-Asberg Depression Rating Scales (MADRS), Hospital Anxiety and Depression Scale (HADS), Global Assessment of Functioning (GAF), quality of life (RAND-36), Kimberley Indigenous Cognitive Assessment of Depression (KICA-dep), Toronto Alexithymia Scale–20 (TAS-20), Clinical Dementia Rating (CDR), Mini-Mental State Examination (MMSE), and Brain-Derived Neurotrophic Factor (BDNF). Participants were assessed at the start of the project and after the sixth month. The research showed that playing Go can improve the quality of life of patients by reducing depression and the severity of AD by increasing the concentration of BDNF [23].

Another method that has proven to positively affect cognitive and functional abilities in the elderly is functional task exercises (FcTSim). The conducted research showed that 13 one-hour classes over a 10-week period resulted in significant positive changes in the group of people exercising with this method. Importantly, patients were examined before starting the program, after completing the cycle, and 6 months after the end of the program. It should be emphasized that the positive effects continued after six months. As part of the patient assessment, a significant number of research tools were also used: Neurobehavioral Cognitive Status Examination (NCSE), Version Verbal Learning Test (CVVLT), Category Verbal Fluency Test (CVFT), Trail Making Test A (TMT-A), Trail Making Test B (TMT-B), Lawton Instrumental Activities of Daily Living Scale (Lawton IADL), and Problems in Everyday Living Test (C-PEDL) [19].

On the other hand, the research by DeokJu Kim analyzed the problem of patients with cognitive disorders. For the purposes of the research, a memory-based occupational therapy program (recollection-based occupational therapy program) was created. As part of the rehabilitation program, five pillars of therapy were distinguished: physical, gardening, musical, artistic, and activities in everyday life (IADL). Participants in this study participated in 24 one-hour lessons five times a week. Participants were tested at the beginning and end of the program using Functional Independent Measure (FIM), Korean Mini-Mental Status Examination (K-MMSE), Subjective Memory Complaints Questionnaire (SMCQ), Short-Form Geriatric Depression Scale-K (SGDS- K), and Geriatric Quality of Life-Dementia (GQOL-D). The results of this study showed a positive effect of the application of the recollection-based occupational therapy program in people with cognitive impairment [20].

The study selected for the above review is also a multicentre study made in Finland. This study was conducted over a period of 24 months to see if any additional factors predisposed cognitive disorder. Its participants were instructed in areas important to health to prevent cognitive impairment. As part of this intervention, training was carried out in the field of diet, physical activity, mental training, and the proper management of the cardiovascular system. The information provided was to be implemented by the participants in their daily lives. Additionally, they were encouraged to seek professional help in these areas. Participants were tested at the start of the program, at 6, 12, and 24 months. As part of the assessment, data such as age, gender, education, annual income, cognitive abilities (MMSE score), blood pressure, BMI, total cholesterol, broken down into fractions, and glucose concentration were taken into account. The results of these large-scale studies did not show that any additional factors influenced the incidence of cognitive disorders, confirming the positive effect of healthy habits in these areas [29].

An alternative approach to the treatment of Alzheimer’s disease was presented by the American researchers Nicholas R. Simmons-Stern, and Rebecca G. Deason, et al. For their clinical trial, they used 12 elderly people diagnosed with AD, and 12 elderly people who were excluded from cognitive impairment. As part of the music therapy, 80 four-line songs and poems for children were used, the content of which was changed to include everyday objects and activities (IADL) that an elderly person should independently perform, e.g., taking medications. All these simple and short songs and rhymes were recorded by a professional singer in a studio. All recordings were carefully systematized in terms of duration, pace, and number of words. Each participant was examined individually. During the research session, the recordings were played three times with the text displayed to them, which took about 1.5 h. Then, the study participants were asked about the activities or objects contained in the lyrics of songs or poems. The above studies showed that information conveyed in the form of simple songs was better remembered than in the form of rhymes. This result was reflected both in people with AD and in elderly people without cognitive impairment [24].

The Greek scientists Aligizakis Eftychios, Sivaropoulos Nektarios, and Gryllaki Nikoleta also took up music therapy. In their 30-month study, 31 people with mild to severe AD were taken. The Mini-Mental State Examination Questionnaire (MMSE) was used for clinical evaluation of the study results. The test was performed every 6 months. The main part of the therapy was divided into four sections: listening to music, songs, improvisation, and talking. Classes were held three times a week and lasted no longer than one hour. During the therapy session, participants dealt with various musical instruments. During the classes, they not only listened to the playing of instruments but also played them themselves. These activities emphasized interactive participation. The results of this study showed that this active form of music therapy improved interaction with AD patients, it also had a positive effect on the stage of the disease, and improved the communication skills and emotional state of the patient [25].

Similarly, Chinese researchers have studied the effects of music therapy on patients with Alzheimer’s disease. Their study included 9 elderly people diagnosed with mild or moderate AD in a care center in Wuhan. The average age of the respondents was 80 years. In order to maintain the sense of safety of the participants, the tests took place at the center. Subjects were tested at the beginning and end of the project with the Autobiographical Memory Test (AMT) and the Geriatric Depression Scale (GDS-15). Based on the data collected during the interview, each participant was individually adjusted to a therapeutic program. Each of the 16 treatment sessions lasted 45 min and took place twice a week. During the intervention, the therapist and the assistant constantly monitored the subject in terms of responses and their behavior. Then, after deliberation, they agreed on changes for the next session. This study showed a positive effect of music therapy in patients with AD, which was reflected in the results of the AMT and GDS-15 tests, after the use of the therapy [26].

In the recent discussed studies, the authors focused on the improvement of cognitive functions and reduction of disability in patients with Alzheimer’s disease.

For an effective therapy of a patient suffering from Alzheimer’s disease, their ability to remember seems very important. We distinguished between procedural memory (consisting of the accumulation of motor experiences, and ways of behavior such as writing or playing tennis) and declarative memory (acquired in the learning process) [30,31]. Already at the beginning of the disease, learning ability is impaired, although declarative memory is not yet impaired [32,33]. Only at a later stage, does the patient not remember exercises, commands to exercise, etc. It is very important, therefore, to learn motor skills at an early stage of the disease, because at this stage, memorizing by learning (declarative memory) is possible and should contain all the elements necessary for functioning at a later stage of the disease [33]. Teaching simple verbal commands and movements that build basic human activities find neurophysiological justification, because axonal transport, activated by the work of muscles, is also declarative [34,35]. It is a condition of the plasticity of the nervous system and the plasticity of the muscles [36,37,38]. This is confirmed by the research by Hirono N. et. al. Their results showed that patients with mild AD can acquire motor, perceptual and cognitive skills, and that the nervous system supporting procedural skills is not related to the neural systems responsible for declarative memory [39].

### 4.1. Research Value

Our systematic review justifies various forms of activity to improve the functional and cognitive abilities of people with Alzheimer’s disease in different stages of the disease.

### 4.2. Study Limitation

This systematic review has several limitations. The aim of the work was to determine the influence of functional exercise on the condition of patients in various stages of Alzheimer’s disease, but we found a lot of work focusing on the mild stages and only several on the moderate and advanced stages of the disease. Probably because this stage is associated with significant mobility limitations, as in any other aspect of the analyzed functions in this disease entity. Therefore, it seems very important to look for opportunities for functional improvement in the advanced stages of Alzheimer’s disease. On the other hand, articles presenting studies on a group of moderate-stage patients did not meet the inclusion criteria for our systematic review. Moreover, the selection of literature was limited to articles in English. Therefore, it is possible that we may have omitted research in other languages.

## 5. Conclusions

This systematic review revealed a beneficial effect in improving cognitive and motor functions after the application of various kinds of activities, especially in the early and mild stages of Alzheimer’s disease.

Improvement followed all the types of physical interventions analyzed.

## Figures and Tables

**Figure 1 ijerph-19-11769-f001:**
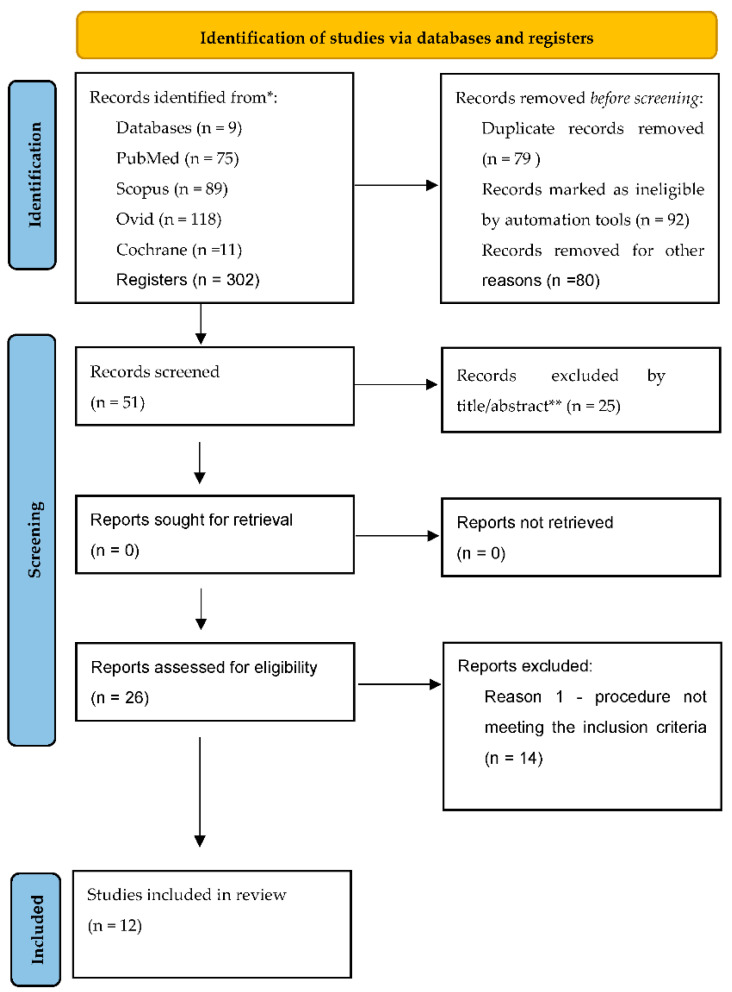
PRISMA flow diagram.

**Table 1 ijerph-19-11769-t001:** Characteristics of the studies included in the review.

Author	Sample (N)	Intervention	Duration, Frequency, and Exercise Intensity
Morris et al. [15]	AEx = 39, ST = 37 older adults over 55 early stage of AD	AEx: aerobic exercise ST: non-aerobic (stretching and toning control program)	26 weeks 150 min per week/3–5 sessions HR from 40–55% to 60–75%
Sobol et al. [16]	IG = 29, CG = 26 age 52–83 years mild stage of AD	IG/CG: aerobic exercises (rgometer bicycle, cross trainer, treadmill), Non-aerobic (strength training of the lower extremity)	IG/CG: 16 weeks 1 h 3 times weekly moderate-to-high intensity The following 12 weeks comprised 10 min of warm up followed by 3 times 10 min of moderate-to high intensity with small breaks of 2–5 min
Vidoni et al. [17]	AEx = 33, ST = 32 adults over 55 early stage of AD	AEx: aerobic exercise; ST: non-aerobic (stretching and toning control program, core strengthening, resistance bands, modified Tai Chi, modified yoga)	AEx: 26 weeks from 60 min a week, increasing the weekly exercise time by about 21 min a week, reached 150 min a week, over 3–5 sessions. ST: keep HR below 100 beats per min.
Lam et al. [18]	IG = 171, CG = 218 65 years old mild cognitive impairment	IG: 24-style Tai Chi CG: stretching and relaxation exercises	12 months ≥30 min per day ≥3 days per week intensity not specified
Law et al. [19]	IG = 43, AC = 40 older adults over 60 mild cognitive impairment	IG—FcTSim (program facilitated by an occupational therapist) AC—Computer-based cognitive training (visual searching, forwardbackward digit recall and calculation) Cognitive strategy training	IG: 10 weeks 13 sessions till 40 to 50 min AC: over 10 weeks, 6 sessions Total 60 min intensity not specified
Kim [20]	EG = 18, CG = 17 older adults mild stage of AD	EG: cognitive stimulus program based on recollection (physical, horticultural, musical, art, IADL activity) CG: activities at centers (physical activity, recreation, watching TV)	EG: 24 sessions 5 times per week 60 min per session Easy, medium to hard CG—regular activities intensity not specified
Parra et al. [21]	EG = 171, CG = 163 adults over 65 did not report stage	EG/CG—DAT Occupational therapy Physical therapy Cognitive therapy (psychology, socio-cultural animation, and complementary therapies)	EG: 8 months every day, 1 or 2 a week weekly sessions of 45 min intensity not specified
Yu et al. [22]	AD = 10, older adults over 65, mild-to-moderate stage AD	AEx: aerobic exercise (cycling), focus group,	EG: 6 months, 3 times a week, session of 45 min
Lin et al. [23]	AD = 147 (CG = 49, SGGI = 49 LGGI = 49) Older adults, did not report stage	Playing Go-game	SGGI—1 h daily for 6 months LGGI—2 h dayly for 6 months CG—non playing
Simmons-Sterna et al. [24]	AD/EG = 12, CG = 12 Older adults, did not report stage	EG/CG Musicaltherapy, simple songs and spoken recorgings	EG: 1.5 h, one session intensity not specified
Eftychios et al. [25]	AD/EG = 31, Older adults, Mild, moderate, severe stage of AD	Musicaltherapy, playing on instruments, singing, litening	EG: 30 months 1 h, three times a week,
Zhang et al. [26]	AD/EG = 9, Older adults, Mild and moderate stage of AD	Musicaltherapy, music listenig	EG: 45 min of 16 sessions, Twice a week

**Legend:** AD—Alzheimer disease; AEx = aerobic exercise group; ST = stretching and toning control group; IG—intervention group; EG—experimental group; CG—control group; FcTSim—functional task exercise; AC—active control group; DAT—therapy with dogs based on the Comprehensive Cognitive Activation Program in Dementia; HR—Heart rate; SGGI—Short Go Game Intervention, LGGI—Long Go Game Intervention.

**Table 2 ijerph-19-11769-t002:** Instruments used and results.

Author	Type of Intervention	Instruments	Outcomes	Effect of Exercise
Morris et al. [15]	Aerobic exercise Non-aerobic exercise	DAD, CSDD, CPX, MRI	Memory, executive function composite scores, functional ability, depressive symptoms, peak VO2, brain volumes	Positive
Sobol et al. [16]	Aerobic exercises Non-aerobic exercise	CEPT, MMSE, SDMT, NPI	VO2 peak, Mental speed Attention, Neuropsychiatric symptoms,	Positive
Vidoni et al. [17]	Aerobic exercise Non-aerobic exercise	DAD, RUD-Lite	Functional dependence, Informal caregiver time required, Cognition	Positive
Lam et al. [18]	Tai Chi Stretching and relaxation exercises	CDR, DAD, CSDD, NPI, BBS	Cognitive functions, Neuropsychiatric assessments, Depression, body balance	Positive
Law et al. [19]	Program facilitated by an occupational therapist Computer-based cognitive training Cognitive strategy training	NCSE, CVVLT, CVFT, TMT-A, TMT-B	Cognitive functions, Lawton IADLC-PEDL	Positive
Kim [20]	Cognitive stimulus program based on recollection Regular activities at centers	FIM, K-MMSE, SMCQ, SGDS-K, GQOL-D	Functional independence, Mental speed, Memory, Depression level, Quality of Life	Positive
Parra et al. [21]	DAT Occupational therapy Physical therapy Cognitive therapy	MEC, Mdified Barthel Index, CSDD, NPI	Mental speed, Motor skills, Depression level, Neuropsychiatric assessments,	Positive
Yu et al. [22]	Aerobic exercise (cycling) Focus group	MMSE, CDR, Borg Rating of Perceived Exertion Scale, Heart Rate Reserve	Mental speed, Memory, Overall functioning, Stress reduction, Improvement of attitude	Positive
Lin et al. [23]	Playing Go-game	MADRS, HADS, GAF, RAND-36, KICA-dep, TAS-20, CDR, MMSE, BDNF	Quality of life, Depression level, Mental speed, Memory, Protein Levels of BDNF	Positive
Simmons-Sterna et al. [24]	Listening to recorded songs, and spoken recordings	MMSE, MOCA	Mental speed, Memory enhancement	Positive
Eftychios et al. [25]	Music listening, playing on instruments	MMSE	Mental speed, Memory, Increase the life quality of the participants	Positive
Zhang et al. [26]	Music listening	AMT, GDS-15	Memory enhancement	Positive

**Legend:** DAD—Disability Assessment of Dementia; CSDD—Cornell Scale for Depression in Dementia; CPX—cardiopulmonary exercise test; MRI = magnetic resonance imagery; CEPT—Test of maximal oxygen uptake; MMSE—Mini-Mental Status Examination; SDMT—Symbol Digit Modalities Test; NPI—Neuropsychiatric Inventory; RUD-Lite—Resource Utilization for Dementia Lite Scale; BBS—Berg Balance Scale; NCSE—Chinese version of Neurobehavioral Cognitive Status Examination; CVVLT—Chinese version of Verbal Learning Test; CVFT—Category Verbal Fluency Test; TMT-A -Trail Making Test A; TMT-B—Chinese version Trail Making Test B; C-PEDL—Problems in Everyday Living Test; FIM—Functional Independent Measure; K-MMSE—Korean Mini-Mental Status Examination; SMCQ—Subjective Memory Complaints Questionnaire; SGDS-K—Short-Form Geriatric Depression Scale-K; GQOL-D—Geriatric Quality of Life-Dementia; MEC—Mini-Cognitive State Examination; Lawton IADL—Chinese versions of Lawton of Instrumental Activities of Daily Living Scale; MADRS—Montgomery-Asberg Depression Rating Scales, HADS—Hospital Anxiety and Depression Scale, GAF—Global Assessment of Functioning, RAND-36—quality of life test, KICA-dep-Kimberley Indigenous Cognitive Assessment of Depression, TAS-20—Toronto Alexithymia Scale–20, CDR—Clinical Dementia Rating, BDNF—level of Brain Derived Neurotrophic Factor, MOCA—Montreal Cognitive Assessment, AMT—Autobiographical Memory Test, GDS-15—Geriatric Depression Scale.

**Table 3 ijerph-19-11769-t003:** Evaluation of the methodological quality of the studies included.

	Sample Randomization	Random Sequence Generation	Blinding of Evaluators	Similar Groups in the Initial Assessment	Inclusion Criteria for Participants	Description of the Experimental Protocol	Statistical Comparison among Groups	Description of Sample Losses	Results Description
Morris et al. [15]	YES	YES	YES	YES	YES	YES	YES	YES	YES
Sobol et al. [16]	YES	YES	YES	YES	YES	YES	YES	YES	YES
Vidoni et al. [17]	YES	NO	YES	YES	YES	YES	YES	NO	YES
Lam et al. [18]	YES	NO	YES	YES	YES	YES	YES	NO	YES
Law et al. [19]	YES	YES	YES	YES	YES	YES	YES	YES	YES
Kim et al. [20]	YES	NO	NO	YES	YES	NO	YES	NO	YES
Parra et al. [21]	YES	NO	NO	YES	YES	YES	YES	YES	YES
Yu et al. [22]	YES	NO	NO	YES	YES	YES	NO	NO	YES
Lin et al. [23]	YES	NO	YES	YES	YES	YES	YES	NO	YES
Simmons-Sterna et al. [24]	YES	NO	YES	YES	YES	YES	YES	YES	YES
Eftychios et al. [25]	YES	NO	NO	YES	YES	YES	YES	YES	YES
Zhang et al. [26]	YES	NO	NO	YES	YES	YES	YES	NO	YES
Statistical analysis according to assessment item n/N(%)	12/12 (100%)	3/12 (25%)	7/12 (58%)	12/12 (100%)	12/12 (100%)	11/12 (92%)	11/12 (92%)	6/12 (50%)	12/12 (100%)

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
