# Peer review of "Functional Therapeutic Strategies Used in Different Stages of Alzheimer’s Disease—A Systematic Review"

_ijerph, 2022, doi:10.3390/ijerph191811769_

Round 1
Reviewer 1 Report
This article can be published with minor revision of the functional therapeutic strategies used in different stages of Alzheimer's disease.At the beginning comes treatment with music. In the 1500s, Ottoman physicians realized that this was the case.Ottoman physicians in the 1500s realized that this was the case.they did. In the clinics called DarüÅŸÅŸifa, he tried to treat these groups of patients with music in different positions.rdir. Alzheimer's disease and music in todayThere are dozens of articles on the k relationshipThis article can be published with minor revision of the functional therapeutic strategies used in different stages of Alzheimer's disease.At the beginning comes treatment with music. In the 1500s, Ottoman physicians realized that this was the case.Ottoman physicians in the 1500s realized that this was the case.they did. In the clinics called DarüÅŸÅŸifa, he tried to treat these groups of patients with music in different positions.rdir. Alzheimer's disease and music in todayThere are dozens of articles on the k relationshipThis article can be published with minor revision of the functional therapeutic strategies used in different stages of Alzheimer's disease.At the beginning comes treatment with music. In the 1500s, Ottoman physicians realized that this was the case.Ottoman physicians in the 1500s realized that this was the case.they did. In the clinics called DarüÅŸÅŸifa, he tried to treat these groups of patients with music in different positions.rdir. Alzheimer's disease and music in todayThere are dozens of articles on the relationship.In the sense of contributing to the author, "Current I recommend him to refer to the article Doi.:10.18863/pgy.238188 published in Approaches in Psyhiatry, 2016.
Author Response
Manuscript ID: ijerph-1875189
Type of manuscript: Article
Title: Functional therapeutic strategies used in different stages of Alzheimer’s Disease – a systematic review.
Dear Reviewers,
Thank you very much for the analysis of our manuscript. We really appreciate your comments and indication of fragments that should be corrected and explained. Considering your suggestions, all mistakes were corrected. The introduction of corrections and changes in the text could have caused the numbering of the lines to shift. In order to avoid misunderstandings, changes introduced in the text are marked in blue and additionally, the manuscript was sent in the change tracking mode.
Reviewer #1:
Thank you very much for the very quick and thorough analysis of our manuscript.
Regarding the remarks and comments, I am writing back.
The following comments and answers:
Comments and Suggestions for Authors
This article can be published with minor revision of the functional therapeutic strategies used in different stages of Alzheimer's disease.At the beginning comes treatment with music. In the 1500s, Ottoman physicians realized that this was the case.Ottoman physicians in the 1500s realized that this was the case.they did. In the clinics called DarüÅŸÅŸifa, he tried to treat these groups of patients with music in different positions.rdir. Alzheimer's disease and music in todayThere are dozens of articles on the k relationshipThis article can be published with minor revision of the functional therapeutic strategies used in different stages of Alzheimer's disease.At the beginning comes treatment with music. In the 1500s, Ottoman physicians realized that this was the case.Ottoman physicians in the 1500s realized that this was the case.they did. In the clinics called DarüÅŸÅŸifa, he tried to treat these groups of patients with music in different positions.rdir. Alzheimer's disease and music in todayThere are dozens of articles on the k relationshipThis article can be published with minor revision of the functional therapeutic strategies used in different stages of Alzheimer's disease.At the beginning comes treatment with music. In the 1500s, Ottoman physicians realized that this was the case.Ottoman physicians in the 1500s realized that this was the case.they did. In the clinics called DarüÅŸÅŸifa, he tried to treat these groups of patients with music in different positions.rdir. Alzheimer's disease and music in todayThere are dozens of articles on the relationship.In the sense of contributing to the author, "Current I recommend him to refer to the article Doi.:10.18863/pgy.238188 published in Approaches in Psyhiatry, 2016.
Thank you very much for your appreciation of our manuscript.
Taking into account your pertinent comments, we reviewed the treatment strategies again to include music therapy and extended the time frame back to 10 years. In this way, we have included another three articles on music therapy, another on aerobic exercise, and the use of Go in patients with Alzheimer's disease. We expanded the overview with another 5 articles, bringing the total to 12 manuscripts that meet the inclusion criteria. The attached articles rank among the references 25 to 29:
- Fang Yu, Ruth M. Swartwood, Feasibility and Perception of the Impact From Aerobic Exercise in Older Adults With Alzheimer’s Disease. American Journal of Alzheimer’s Disease & Other Dementias® 27(6) 397-405. DOI: 10.1177/1533317512453492.
- Lin Q, Cao Y, Gao J. The impacts of a GO-game (Chinese chess) intervention on Alzheimer disease in a Northeast Chinese population. Front. Aging Neurosci. 2015 7:163. doi: 10.3389/fnagi.2015.00163.
- Nicholas R. Simmons-Sterna, Rebecca G. Deasona, Brian J. Brandlera,b, Bruno S. Frustacea,b, Maureen K. O’Connora, , Brandon A. Allyd, Andrew E. Budsona. Music-Based Memory Enhancement in Alzheimer’s Disease: Promise and Limitations. Neuropsychologia. 2012; 50(14): 3295–3303. doi:10.1016/j.neuropsychologia.2012.09.019.
- Eftychios, A., Nektarios, S. and Nikoleta, G. (2021) Alzheimer Disease and Music-Therapy: An Interesting Therapeutic Chal-lenge and Proposal. Advances in Alzheimer’s Disease, 10, 1-18. doi.org/10.4236/aad.2021.101001.
- Anshuang Zhang, Yunpeng Yang, Ming Xu. Clinical Observation of Computer Vision Technology Combined with Music Therapy in the Treatment of Alzheimer’s Disease. Emergency Medicine International Volume 2022, Article ID 2567340, 12 pages, doi.org/10.1155/2022/2567340.
If anything still needs to be improved, please point out the shortcomings.
Thank you very much for the suggestion.
Thank you very much for your time.
Reviewer 2 Report
The manuscript entitle Functional therapeutic strategies used in different stages of Alzheimer’s Disease – a systematic review by Anna Olczak et al. describes the benefits of several motor activity in order to reduce Alzheimer progress in the illness patients. The manuscript is well organized, and methods are well described. My main concern is the impact of this paper. Only seven papers were revised and included in the present study, even though they analyzed more than two hundred. The scarce number of the research article analyzed reduces the impact of the study. In fact, they only searched for the last five years. If the authors consider it, they could expand the number of the years. This could help to make the study stronger and therefore, more attractive to the reader. Manuscript includes only some results of the several papers analyzed, but this reviewer consider that a revision paper should include a general idea common to the analyzed studies, and not the same idea described in each one.
In addition, authors should revise the text. I consider that an English native could help to get better it. I have read some gramatical errors and typo, typical in nonnative people but it does not minimize the value of the study.
Author Response
Manuscript ID: ijerph-1875189
Type of manuscript: Article
Title: Functional therapeutic strategies used in different stages of Alzheimer’s Disease – a systematic review.
Dear Reviewers,
Thank you very much for the analysis of our manuscript. We really appreciate your comments and indication of fragments that should be corrected and explained. Considering your suggestions, all mistakes were corrected. The introduction of corrections and changes in the text could have caused the numbering of the lines to shift. In order to avoid misunderstandings, changes introduced in the text are marked in blue and additionally, the manuscript was sent in the change tracking mode.
Reviewer #2:
Thank you very much for the very quick and thorough analysis of our manuscript.
Regarding the remarks and comments, I am writing back.
The following comments and answers:
Comments and Suggestions for Authors
The manuscript entitle Functional therapeutic strategies used in different stages of Alzheimer’s Disease – a systematic review by Anna Olczak et al. describes the benefits of several motor activity in order to reduce Alzheimer progress in the illness patients. The manuscript is well organized, and methods are well described. My main concern is the impact of this paper. Only seven papers were revised and included in the present study, even though they analyzed more than two hundred. The scarce number of the research article analyzed reduces the impact of the study. In fact, they only searched for the last five years. If the authors consider it, they could expand the number of the years. This could help to make the study stronger and therefore, more attractive to the reader. Manuscript includes only some results of the several papers analyzed, but this reviewer consider that a revision paper should include a general idea common to the analyzed studies, and not the same idea described in each one.
Thank you very much for your appreciation of our manuscript.
Taking into account your pertinent comments, we reviewed the treatment strategies again to include music therapy and extended the time frame back to 10 years. In this way, we have included another three articles on music therapy, another on aerobic exercise, and the use of Go in patients with Alzheimer's disease. We expanded the overview with another 5 articles, bringing the total to 12 manuscripts that meet the inclusion criteria. The attached articles rank among the references 25 to 29:
- Fang Yu, Ruth M. Swartwood, Feasibility and Perception of the Impact From Aerobic Exercise in Older Adults With Alzheimer’s Disease. American Journal of Alzheimer’s Disease & Other Dementias® 27(6) 397-405. DOI: 10.1177/1533317512453492.
- Lin Q, Cao Y, Gao J. The impacts of a GO-game (Chinese chess) intervention on Alzheimer disease in a Northeast Chinese population. Front. Aging Neurosci. 2015 7:163. doi: 10.3389/fnagi.2015.00163.
- Nicholas R. Simmons-Sterna, Rebecca G. Deasona, Brian J. Brandlera,b, Bruno S. Frustacea,b, Maureen K. O’Connora, , Brandon A. Allyd, Andrew E. Budsona. Music-Based Memory Enhancement in Alzheimer’s Disease: Promise and Limitations. Neuropsychologia. 2012; 50(14): 3295–3303. doi:10.1016/j.neuropsychologia.2012.09.019.
- Eftychios, A., Nektarios, S. and Nikoleta, G. (2021) Alzheimer Disease and Music-Therapy: An Interesting Therapeutic Chal-lenge and Proposal. Advances in Alzheimer’s Disease, 10, 1-18. doi.org/10.4236/aad.2021.101001.
- Anshuang Zhang, Yunpeng Yang, Ming Xu. Clinical Observation of Computer Vision Technology Combined with Music Therapy in the Treatment of Alzheimer’s Disease. Emergency Medicine International Volume 2022, Article ID 2567340, 12 pages, doi.org/10.1155/2022/2567340.
Thank you very much for this suggestion.
In addition, authors should revise the text. I consider that an English native could help to get better it. I have read some gramatical errors and typo, typical in nonnative people but it does not minimize the value of the study.
We have made another revision of the text in terms of the correctness of the English language. Hopefully, the improvement is enough.
If there are still any shortcomings, please indicate the place.
Thank you very much for your thorough analysis of our work.
Thank you very much for your time.
Round 2
Reviewer 2 Report
The manuscript has improved and now I consider it could be published if the editor considers it.